# Emergence of Network Motifs in Deep Neural Networks

**DOI:** 10.3390/e22020204

**Published:** 2020-02-11

**Authors:** Matteo Zambra, Amos Maritan, Alberto Testolin

**Affiliations:** 1Department of Civil, Environmental and Architectural Engineering, University of Padova, Via Marzolo 9, 35131 Padova, Italy; 2Department of Physics and Astronomy, University of Padova; Istituto Nazionale di Fisica Nucleare—Sezione di Padova, Via Marzolo 8, 35131 Padova, Italy; amos.maritan@pd.infn.it; 3Department of General Psychology, University of Padova, Via Venezia 8, 35131 Padova, Italy; 4Department of Information Engineering, University of Padova, Via Gradenigo 6/b, 35131 Padova, Italy

**Keywords:** deep learning, artificial neural networks, network motifs, complex systems, 90B10, 94C15, 68Q32, 68T05

## Abstract

Network science can offer fundamental insights into the structural and functional properties of complex systems. For example, it is widely known that neuronal circuits tend to organize into basic functional topological modules, called network motifs. In this article, we show that network science tools can be successfully applied also to the study of artificial neural networks operating according to self-organizing (learning) principles. In particular, we study the emergence of network motifs in multi-layer perceptrons, whose initial connectivity is defined as a stack of fully-connected, bipartite graphs. Simulations show that the final network topology is shaped by learning dynamics, but can be strongly biased by choosing appropriate weight initialization schemes. Overall, our results suggest that non-trivial initialization strategies can make learning more effective by promoting the development of useful network motifs, which are often surprisingly consistent with those observed in general transduction networks.

## 1. Introduction

The topological structure of complex networks can be characterized by a series of well-known features, such as the small-world and scale-free properties, the presence of cliques and cycles, modularity, and so on, which are instead missing in random networks [1,2,3,4,5]. It has been shown that another distinguishing feature is the presence of so-called network motifs [6], which are recurring patterns of interconnections that might serve as building blocks for the evolution of more complex functional units [7,8]. One might thus hope to “understand the dynamics of the entire network based on the dynamics of the individual building blocks” (see Chapter 3 in Reference [9]). In this respect, we can regard network motifs as basic structural modules which bear (in a topological sense) meaningful insights about the holistic behavior of the system as a whole.

Here we apply this perspective to the study of multi-layer (deep) neural networks, which are one of the most popular frameworks used in modern artificial intelligence applications [10,11]. Despite the impressive performance achieved by deep networks in challenging cognitive tasks, such as image classification [12], automatic machine translation [13] and discovery of sophisticated game strategies [14], such systems are still poorly understood [15]. To quote Reference [16], “the theoretical principles governing how even simple artificial neural networks extract semantic knowledge from their ongoing stream of experience, embed this knowledge in their synaptic weights, and use these weights to perform inductive generalization, remains obscure”. The inscrutability of deep learning models mostly stems from the fact that their behavior is the result of the non-linear interaction between many elements, which motivates the use of network science techniques to reveal emergent topological properties [17].

The primary question we address in the present work is whether is it possible to observe the emergence of well-defined network motifs even if the initial (between layer) topology corresponds to a fully-connected graph, where each node (neuron) is connected to all nodes in the neighboring layers. The underlying assumption is that some traces of the learning dynamics will be nevertheless recorded in the final model topology in the form of basic functional modules, thus opening the possibility to relate local network properties with the functioning of the system as a whole. Furthermore, given that the objective of deep learning is to extract high-order statistical features from the data distribution [18], we ask to what extent the final topology depends on intrinsic properties of the training data. To this aim, we systematically compare the motifs emerging in a deep network trained on two different synthetic environments, created according to different generative models.

## 2. Methods

### 2.1. Neural Network Architectures

A simple multi-layer feed-forward network was built and trained using the Keras deep learning framework (See https://keras.io/ for documentation). The external deep learning libraries and motifs mining software were used as provided, while the rest of the system was coded from scratch in Python 3.7.4 (See https://docs.python.org/release/3.7.4/ for documentation). The Appendix A (source codes) are available at https://github.com/MatteoZambra/SM_ML__MScThesis. In order to better assess the robustness of our simulations, three different neural network architectures with varying number of neurons have been considered (see Table 1). In all the cases, the architecture consists of a fully-connected multi-layer perceptron model. A broader overview on the robustness results is given in Appendix B, where we also discuss simulations related to a larger-scale, more realistic machine vision problem (e.g., classification of handwritten digits).

### 2.2. Learning Environments

Inspired by the recent study of Saxe and colleagues [16], two synthetic sets of data were purposefully generated to embed different statistical structures, to investigate whether different environmental conditions would lead to the emergence of specific topological signatures. The first environment, encoded as a binary tree data set, contained a hierarchical structure. The second environment, encoded as an independent clusters data set, resolved in a more sparse and glassy statistical footprint, as shown in Figure 1. The structure of each data set will be briefly reviewed in the following; for a more detailed description, the reader is referred to the Appendix C. As customary in the machine learning literature, a data instance is thought of as a vector of random variables representing some measurable features. We can conceive each data instance as being generated by sampling from a (probabilistic) graphical model, whose nodes represent the random variables of interest, and whose edges represent their mutual relationships. For both the data sets considered, samples can be divided into four classes, as explained more in detail below.

#### 2.2.1. Binary Tree Data Set

The binary tree data set is thought as a slight modification of the generative model illustrated in Reference [16]. This data generator is designed to create data instances displaying a hierarchical structure, as shown in Figure 1a. Each generator run produces a data instance, whose features attain values among {−1,+1}. The initial feature (root node) is sampled uniformly and its value diffuses through the tree branches. The rationale underlaid is as follows: If the root node attains the value +1, then the left child inherits the +1 value and the right child (together with all its progeny) is assigned the value −1. Contrarily, if the root node happens to be −1, then the right child inherits the +1 value. From the root node children on, the criterion is probabilistic. One of the children of a +1 node inherits the same value with probability ε, that is a threshold set *a priori* to 0.3 (see Figure 2a). Clearly, the smaller this threshold, less likely is the value to flip. To perform such a stochastic flip, for each node one samples a value *p* uniformly distributed in [0,1]. For example: a node *i* has the value +1. Given ε, sample p∼U(0,1). If p>ε, then the left child of *i* (which has the node index equal to 2×i+1, where the root node number is (Note that the node number is not the value stored in such node. This latter may be −1 or +1 while the former ranges from 0 to the total number of nodes $N=2^{D}-1$. D=1,… refers instead to the depth of the tree, that is, how many node levels it has. Note that a linear array storage is used for the binary tree data structure). i=0) inherits the value +1 and the right child instead inherits −1. If p≤ε, the left child inherits −1 and the right child inherits +1. The features of each data sample are the collection of all the nodes in the tree structure for a given generator run, with the first feature being the root node, the right and the left children contain the second and third features respectively and so forth.

The number of classes is set selecting a level in the tree: The root node (level 1) identifies two axes of distinction, that is whether its value is +1 or −1. The data set fed to the network for our analyses is generated accounting for a level of distinction set to 2, which thus generates four different classes (see Appendix C). The choice of the level of detail set to 2 means that one accounts for the classes identified by the equality of the data samples up to the random variables of the second tree level plus the respective outcomes; all the other variables may differ, thus guaranteeing some variability in the patterns belonging to the same class.

#### 2.2.2. Independent Clusters Data Set

The independent clusters data set is designed to endow the data instances ensemble with a block-diagonal statistical signature, as displayed in Figure 1b. For such purpose, a simple graph is created as in Figure 3a. For consistency with the number of features and classes of the binary tree data set, the same number of random variables involved is chosen, and the number of independent groups is set according to the number of classes of the binary tree data set. In the full graph, as the embedding shown in Figure 3a, some connections are gradually eliminated to reproduce the situation in Figure 3b. The rationale beneath is as follows: In a fashion similar to the simulated annealing algorithm [19], a temperature schedule is set, where the initial temperature is chosen to be the reciprocal of the longest edge and similarly the final value is the reciprocal of the shortest edge
(1)$$\left\{\begin{array}{ccc} T_{0} & = & {\left(\max_{e}\left\{e\in E\right\}\right)}^{-1}\\ T_{f} & = & {\left(\min_{e}\left\{e\in E\right\}\right)}^{-1}. \end{array}\right.$$

The schedule steps are equally spaced and the number of such steps is a parameter that has been fine-tuned to have the desired final scenario of disjoint, arbitrarily intra-connected groups. Note that by thus doing it could be that from a step to the next, shorter links may be erased while retaining longer ones, which were short enough to survive in the last step, hence melting schedule must be set sensibly. Unlike simulated annealing, the temperature rises in this case. For each step, edges greater or equal than the inverse of the temperature value are deleted. Each of these four graphs is directed, to allow for more efficient sampling. Each random variable (i.e., node value) is initialized to the value of −1. Subsequently, one selects randomly a group among the four available and the topological order of such nodes is set: The node with no incoming connection has topological order 1. The nearest neighbors of the latter have topological order 2, and so forth (see Figure 2b). For the sake of simplicity, this data set is created in such a way that a random variable of the selected group bears as value its topological order. This translates in a straight-forward classification task since all the data samples belonging to a class are equal: All the features of a data sample are set to −1, except those corresponding to the nodes of the selected group, which are set equal to their topological ordering (see Appendix C for a thorough explanation). The construction of the structure depicted in Figure 3b is motivated by the necessity to dispose of a data set in which some random variables share a probabilistic relationship and some others are independent. Referring to Figure 1b, the regions in the covariance matrix related to dependent variables are visible as the diagonal blocks, while the background appears more *glassy*. This latter is not fully homogeneous but has rather a chessboard-like textured since the numerical values attained by the non-chosen groups will somehow be related to the values of the chosen group.

### 2.3. Initial Conditions

Besides varying the statistical structure of the learning environment, we also investigated whether the emergence of different topological signatures could also be related to the use of different initialization schemes for the connection weights. To this aim, we considered three different initialization schemes. In the first case, we used the classic “Normal” initialization method, where each connection weight is randomly sampled from a Gaussian distribution with zero mean and small variance, that is w∼N(0.0,0.1). In the second case, we exploited the “Orthogonal” initialization method proposed in Reference [20], where weight matrices of adjacent layers are constrained to be orthogonal. This scheme has been proved to grant depth-independent training speed, which is desirable as the network becomes deeper. Finally, in the third case we exploited the popular “Xavier” (equivalently called “Glorot”) initialization method [21], where the mean is zero and the variance of the Gaussian distribution is defined according to the number of connections of each layer, that is:(2)$$\sigma^{2}=\frac{k}{n_{\mathrm{in}}+n_{\mathrm{out}}},$$
where *k* depends on the activation non-linearity and $n_{\mathrm{in},\;\mathrm{out}}$ are the number of incoming/outgoing connections of each layer. This initialization scheme enjoys widespread popularity, since it has been empirically shown to mitigate optimization issues affecting deep networks.

### 2.4. Task and Learning Dynamics

The task accomplished by the network is multi-class classification. The data sets are homogeneous in terms of design matrix dimensions and items in the labels set, thus the network architecture is the same in both the cases: Once the parameters are initialized, the data structure containing these parameters and connectivity information is trained both on the binary tree and independent clusters environments.

Stochastic (mini-batch) gradient descent was used to adjust the network’s weights. Learning rate was initially set to 0.01, and then decayed using a factor of $10^{-6}$. Nesterov acceleration was added with momentum set to 0.6. Given the straightforward structure embedded in our data sets, simple models should be capable of reaching a perfect level of classification accuracy.

### 2.5. Mining Network Motifs

Network motifs can be defined as “patterns of interconnection occurring in complex networks at numbers that are significantly higher than those in randomized networks” [6]. In the present work, arrangements of four and five nodes are inspected. The statistical significance of such a pattern of connections can be identified by computing the *Z*-score (equivalently referred to as significance score), which is the number of times a given motif appears in the considered network with respect to the average number of occurrences of the same motif in an ensemble of random replicas of the original network. It is measured as a distance in units of standard deviations:(3)$$Z=\frac{N_{\mathrm{real}}-<N_{\mathrm{random}}>}{\sigma_{\mathrm{random}}}.$$

Once the neural networks were trained, model parameters were extracted and transposed in a proper graph data structure, so that network motifs mining can be carried on using an external tool. The FANMOD (See http://theinf1.informatik.uni-jena.de/motifs/ for executable, sources, license and relevant papers.) motifs mining software was used to analyse the graph extracted from the model [22]. A comprehensive account on the algorithmic complexity and technical details about network motifs mining tools is beyond the scope of this work. However, a detailed overview of the underlying algorithmic machinery for the FANMOD software is provided in Reference [23], and state-of-the-art advances and performance benchmarks are discussed in Reference [24].

Since retaining all the connection weights of the initial and trained models would imply the presence of redundant network motifs, all weights relatively close to zero were neglected in the subsequent analyses. The identification of negligible weights was carried out by fitting a Normal distribution to the weights histogram, and selecting as exclusion zone the set of the weights $w:p\left(w\right)\geq c\;\max_{w}p\left(w\right)$, where *p* is the fitted Normal density function. The value of the cut-off threshold *c* plays an important role: the greater this value, the smaller the exclusion region (see Appendix D for details). The majority of the connection weights fell in this zone, hence the subsequent analyses accounted for the strongest connections, either positive or negative valued, thus letting to emerge only the most significant topological features.

### 2.6. Biological Analogy: Neurons and Protein Kinases

The internal working of transduction networks is based on the cooperation between processing units, and the subsequent arrangement of those [25] (To deepen the contents of this section, the interested reader is also referred to Reference [9], to which the topic and notation adopted in the following are inspired.). Sensing environmental stimuli, processing this information and eventually transcribing it to gene expression is done by passing this signal through a network whose units are protein kinases. Such units play the role of nodes in the network, and interactions among those—for example, phosphorylation—are the edges. The activity of these units are modelled through first-order kinetics. The essential items are:the kinases of a first layer, the concentration of which is denoted as $X_{j}$, with j=1,…,n;the target kinase of a second layer, the concentration of which is denoted *Y*;the rate of phosphorylation $r\left(Y\right)=Y_{0}\;\sum_{j}v_{j}X_{j}$, being $v_{j}$ the rate of kinase $X_{j}$.

Call $Y_{0}$ and $Y_{p}$ the concentration of un-phosphorylated and phosphorylated kinase *Y* respectively. The concentration of kinase *Y* remains constant, that is $Y_{0}+Y_{p}=Y$. Then the rate of change of activated kinase *Y* is given by the difference between the rate of phosphorylation *r* and the rate of de-phosphorylation of the same kinase *Y*, at a rate α. In formulae:(4)$$\left\{\begin{array}{ccc} r(Y) & = & Y_{0}\;\sum_{j}v_{j}X_{j}\\ Y & = & Y_{0}+Y_{p}\\ \frac{dY_{p}}{dt} & = & r\left(Y\right)-\alpha Y_{p}. \end{array}\right.$$

Referring to the case of steady-state dynamics, straight-forward calculations yield that the concentration of active *Y* is non-linearly proportional to the weighted sum of the inputs $X_{j}$, as depicted in Figure 4:(5)$$Y_{p}=\frac{\sum_{j}w_{j}X_{j}}{1+\sum_{j}w_{j}X_{j}},$$
where $w_{j}=v_{j}\alpha^{-1}$. A sensible threshold value for this weighted sum is thought to be 1 approximatively. After the value of the input exceeds 1, the target kinase activity starts to be sensible. Now assume that this simple model involves *m* target kinases. Then:
(6)$$\sum_{j=1}^{n}\;w_{ij}X_{j}=1\;i=1,\ldots ,m$$
identifies the hyper-plane in the space of the inputs that excludes regions of high and low activity, depending on the connection strengths. It happens that by stacking more of such three-nodes modules (*n* input signals from the kinases $X_{1},\ldots ,X_{n}$ and the target unit *Y*), one can obtain complex geometries of the activity region in the input space. It is shown that the motifs encountered most often in transduction networks are the so-called diamonds and bi-parallel. As we will discuss later, these motifs match those found in our analyses, as depicted below.

The analogy with binary classification is hinged on the creation of the hyper-plane. Assume that the weighted input is $u=\sum_{j}w_{ij}X_{j}$, the numerical value of which is known. To determine whether the unit *Y* is active, one needs to compare the input *u* with the hyper-locus that identifies the regions of activity. Assume that the target unit activates once the threshold 1 is exceeded, then:If u≥1 then target unit *Y* activates and propagated the signal forward in the system to a third layer. ButIf u<1 then *Y* is not sufficiently triggered to propagate the signal, that is, to phosphorylate the next unit.

The hyper-space WX≥1, $1={\left\{1\right\}}^{n}$, identifies the set of weights and activities such that the target units is activated. The subtlety in this analogy is that the weights *W* cover a relevant role too: In transduction networks change of such weights is subjected to regulatory mechanisms or evolutionary pressure [9], and in the process of gene expression transcription these weights values are given. They do not play the role of adjustable parameters in such a way to minimize a given error metric. In neural networks, on the other hand, weights adjustment is pivotal in the learning process, and such variations are performed on a faster timescale.

In neural networks one encounters a similar scenario: A neuron is fed with an array of incoming signals, coming from the activities of the previous layers neurons. The weighted sum of these signals is added to an activation threshold value, called bias. The resulting value undergoes a non-linear transformation. In this way it is possible to identify an hyper-plane in the input space that separates the input patterns of signals, as in Figure 5. The “state equations” of a simple one-hidden-layer network are the following:(7)$$\left\{\begin{array}{ccc} h & = & f\left(xW^{\left(1\right)}+b^{\left(1\right)}\right)\\ y & = & f\left(hW^{\left(2\right)}+b^{\left(2\right)}\right) \end{array}\right.,$$
with
$$\begin{array}{c} x\in R^{N_{\mathrm{input}}},\;h\in R^{N_{\mathrm{hidden}}},\;y\in R^{N_{\mathrm{output}}}\\ b^{\left(1\right)}\in R^{N_{\mathrm{hidden}}},\;b^{\left(2\right)}\in R^{N_{\mathrm{output}}}\\ W^{\left(1\right)}\in R^{N_{\mathrm{input}}\times N_{\mathrm{hidden}}},\;W^{\left(2\right)}\in R^{N_{\mathrm{hidden}}\times N_{\mathrm{output}}}. \end{array}$$

Suppose that this network has one hidden layer with $N_{\mathrm{hidden}}$ units, $N_{\mathrm{input}}$ input units, $N_{\mathrm{output}}$ output units and f(·) is a generic non-linearity. In the framework of neural networks these functions are generally monotonically increasing, as for example the “logistic sigmoid” $\sigma \left(x\right)={\left(1+\exp\left(-x\right)\right)}^{-1}$. The vectors $b^{\left(k\right)}$ represent the activation thresholds of both the hidden units and output units—also called *biases* and the matrices $W^{\left(k\right)}$ are the connection weights, k=1,2. Here the hyper-plane is identified by the weighted sum in the arguments of f(·), which purpose is to capture higher-order correlations in the input features and the composition of many non-linear blocks allows the synthesis of high-level abstraction of the domain [10]. Figure 5 gives a visual idea of the hyper-planes composition and the result in terms of decision boundary geometry. Signal flow in the system, from the input layer units to the output nodes, is strictly feed-forward and once the guessed label is observed in this latter layer, it is compared with the ground truth. Based on the mismatch, model parameters (connection weights and node biases) are adjusted, in such a way to minimize the prediction error, back-propagating such error in a reverse way along the layers constituting the network [26].

Given these analogies between biological transduction networks and artificial neural networks, it is legitimate to hypothesize that information processing in both classes of systems might be carried out using similar computational structures. Reference [9] argues that “Multi-layer perceptrons allow even relatively simple units to perform detailed computations in response to multiple inputs. The deeper one goes into the layers of perceptrons, the more intricate the computations can become”. If one thinks of “intricate computations” as the computation of appropriate decision boundaries, then this task is precisely what is accomplished by multi-layer perceptrons. Individual neurons (absorbing an arbitrary length input) could only discriminate two classes, as in Figure 5a (in this case one has only two input features), but stacking together multiple layers of neurons allows to create more intricate and complex decision loci in the input space, as in Figure 5d. Panels Figure 5b–d refer to the combination of two triads as in Figure 5a, assembled so to form a simple neural network with two input neurons and two output units, with no hidden layers. Figure 5b is the exclusion locus of the triad formed by the input units and the first output unit, Figure 5c analogously refers to the triad in which the output unit involved is the second one. This trivial example shows how hyper-planes designed by simple groups of units arrange to identify less obvious exclusion hyper-subspaces and note that stacking more layers one can go beyond straight lines.

## 3. Results

As we will point out below, our analyses on network motifs displayed an intriguing consistency with transduction networks. Results for the four-nodes motifs will be presented first, since they allow enjoying a broader perspective on the internal functioning of the artificial networks compared to the biological counterparts. Five-nodes motifs allow for a closer inspection of how the learning environment and the emergence of topological structures relate to each other, but at the same time the emerging patterns are less easily interpretable.

### 3.1. Learning Efficacy

For all models the training accuracy peaks to the top value of 1.0 in few epochs. However, as shown in Figure 6, the Normal initialization scheme resulted in the slowest learning convergence. The orthogonal initialization scheme allowed convergence in fewer epochs, while the Xavier scheme resulted in the fastest convergence. These findings suggest that initialization plays a crucial role in shaping learning dynamics: one possible explanation could be that the orthogonal and Xavier schemes impress a sharper fingerprint to the initial significance landscape of network motifs, as we will discuss below. In other words, faster convergence toward the optimal set of connection weights might be promoted by biasing the initial set of network motifs. A sharper initial significance landscape is indeed common in the initialization schemes displaying faster convergence.

### 3.2. Emerging Network Motifs

Figure 7 shows how the weights distribution changes during the course of learning. As noted in previous studies [17] the effect of learning is mostly evident in the tails of the distributions and, in our simulations, especially for the Normal initialization scheme. This suggests that the Orthogonal and Xavier initializations might help building more effective motifs since the beginning, by imposing a stronger bias to the initial structure of the network. A statistical analysis revealed that, indeed, although after learning the weights became larger in absolute value for all the initializations and data sets considered (see Table 2), such difference was significant only for the network initialized using the Normal scheme and trained on the tree data set (p<0.01). Figure 6 depicts how the normal initialization scheme renders a slower convergence. This points out that the correlation between the initial distribution and the evolved distribution in the case of normal initialization decreases, implying also a decreased correlation in the motifs significance profiles, see Figure 8d.

Figure 8 shows the overall trend of change for the most common 4-nodes motifs: The *x*-axis gathers the motifs prototypes, the respective *y* value is the significance score, obtaining a significance profile (Note that significance scores are sometimes normalized, allowing to superimpose (and thus compare) significance profiles referred to different instances of a complex network [27]. However, in the present work normalization is avoided, since the network inspected (hence the size of the system) is the same for all the analyses. Non-normalized scores also allow to better understand the magnitudes of the detected effects.). As evident by comparing Figure 8a–c, the significance profiles resulting from different initialization schemes display a remarkable self-similarity, suggesting that this set of basic structures might support information processing in all the data sets considered. Interestingly, several of such motifs are also consistent with those commonly found in biological transduction networks (the fifth and the ninth motifs from the left), suggesting a potential overlap of computational mechanisms. The fundamental feature of the analogy is the identification of an hyper-plane which classifies the nature of a given input—which comes as a weighted sum, in both transduction and neural networks. As mentioned above, the amount of change in the weights seems to be correlated to the change in the significance profiles—in the case of Normal initialization, the variation is most severe, while in the other two cases the significance profile seem to be already biased before learning takes place.

Figure 9 shows the overall pattern emerging from the analysis of the most common 5-nodes motifs. On the one hand, it is possible to appreciate the exclusiveness of the motifs characterizing each arrangement of learning environments, suggesting that the learning domain influences the emergence of particular topologies. On the other hand, results are less obvious compared to the case of 4-nodes motifs. Referring again to Figure 8a–d, it is not clear the extent in which the emergence of the most significant motifs is spontaneous or biased by the initial profile (black lines). Orthogonal matrices and Xavier schemes, by their design, sample larger parameters values, hence the model configuration is conditioned by the initial, albeit random, weights landscape. Results in Figure 9 do not provide a definitive answer to this question: While it may seem that different network motifs emerge in response to different initial and learning environments, it is not clear why and when a certain topology is observed. The multi-layer perceptron motif (the last but one motif in panel Figure 9a) and its variations appear through the different scenarios. In Reference [9] it is argued that such a structure can be viewed as a combination of diamond four-nodes motifs (the last but one motif in the panels of Figure 8). Albeit one may be tempted to think that motifs emerge as self-organized modules that encode domain-specific information, it is not clear whether the emergence of different motifs stems from the diversity in learning environments and initial conditions, or whether it might partially due to the noisy by-product of the learning dynamics itself.

Interestingly, a very similar patter of findings emerged from an additional set of simulations carried out using a more realistic input (see Appendix B), that is, images of handwritten digits. Although the deep network architecture was fairly different in this case (featuring three hidden layers and a greater number of neurons), the resulting network motifs match those found in the networks trained on the tree and clusters data sets, and also the change in significance profiles follows a comparable trend.

## 4. Discussion

In complex networks, individual units by themselves do not accomplish any particularly relevant function, because it is the coordinated arrangement of groups of units (i.e., their *interactions*) that allows for the emergence of system-level, macroscopic properties [28]. In the present work, we thus explored how information processing in deep networks might emerge as a combination of simple network motifs. Starting from these key observations:larger motifs may be seen as arrangements of smaller motifs, for example “Diamonds combine to form multi-layer perceptron motifs” [25];these smaller motifs arrangement gives rise to more complex computation: “Adding additional layers can produce even more detailed functions in which the output activation region is formed by the intersection of many different regions defined by the different weights of the perceptron” [9];domain representation is carried out by the composition of subsequent non-linear modules, which “transform the representation of one level (starting with the raw input) into a representation at a higher, slightly more abstract level” [10];
we hypothesized that in deep neural networks the learning dynamics may rely on the same processing mechanisms used by transduction networks.

Our simulations suggest that this might indeed be the case—network motifs might form spontaneously for the purpose of efficient information processing, so that each module deals with a small number of input features, and subsequent (deeper) processing can rely on fewer signals from previous neurons. High-level features might thus be abstracted in a layer-wise and motifs-wise fashion.

Notably, our analyses also suggest that some weights initialization strategies give a stronger imprinting to the initial significance landscape of possible motifs. The Normal initialization scheme results in a flatter initial landscape, which might underly the slower convergence speed of this type of initialization. The environment may thus be considered to be learned once relevant information processing structures come to develop—if a scheme provides the initial configuration with a preventive signature of such structures, learning will be much faster.

## 5. Final Remarks and Further Improvements

### 5.1. Evaluation with Other Classes of Deep Learning Models

Our simulations have been focused on feed-forward neural networks, which are the workhorse of deep learning, but there exist many other classes of models to which our methodology could be applied. Notable examples would be deep networks with *bidirectional* [29,30] and *recurrent* [31] connectivity, which might even allow for the emergence of a richer variety of motifs, or models featuring *convolutional* and LSTM architectures [32,33,34]. In this respect, it should be stressed that the proposed approach should hold for the analysis of any network-reliant deep learning model.

### 5.2. Presence of Combinatorial Biases

It would also be useful to better investigate whether particular motifs might emerge simply as a consequence of some combinatorial bias induced by the design topology of the multi-layer perceptron itself. If so, some of the significant four-nodes motifs we detected might appear because of unavoidable initial imprinting due to the topology of the model, and not necessarily because of their critical role in information processing. Further insights into this issue could be gained by also analyzing the case of five-nodes groups, or by enforcing an initial landscape in which the motifs observable are those that appear to be most rare. This way one could observe whether such structures are rejected as a result of learning, favoring instead those discussed above.

### 5.3. Sensitivity to Free Parameters

Referring to the Appendix D, it should be noted that our simulations involved a certain number of parameters (e.g., the threshold used to discretize the weights, or to exclude negligible weights from the analyses) that were often set according to heuristics. However, we should note that the results presented turned out to be robust to small variations in the choice of such parameters. The choice of mining a weighted or unweighted network also plays a role. Here we presented results related to a weighted analysis, which introduces variability in the discovered motifs. More specifically, motifs were composed of connections that fall in four categories: close to zero (i.e., not present), strongly positive, strongly negative and mildly positive/negative. Also, one could either account for most significant or most typical motifs, the former being those instances in the same group of isomorphic graphs that display the largest significance and the latter are those which have a significance score that is closer to the average, over the same isomorphic group. For a broader account on the problem of graph isomorphism, the reader is referred to References [35,36].

### 5.4. Scalability

One important research direction would be to apply the proposed framework for the study of larger-scale deep learning systems, which can learn more intricate statistical structure from big data sets where the patterns might not be easily separable (as in the cases presented here). Scalability remains one of the main concerns and an important direction for further research. This problem does not impact training (which can be optimized using high-performing parallel hardware [37,38]) but rather the motifs mining stage. Motifs searching relies on “edges sampling” [24] and this implies larger computational resources for larger graphs.

### 5.5. Alternatives to Motifs Mining Algorithms

Finally, given that classical motifs search algorithms may present some limitations (e.g., related to the computational complexity, as outlined above), it would be interesting to also investigate approaches based on spectral graph partitioning. Indeed, finding community structure in complex networks is by no means a novel problem (see Reference [39]), and in this respect graph partitioning can be accomplished with spectral techniques. The “spectral” adjective refers to the eigenvalues of the graph Laplacian [40]. However, a difference to pay attention at in this case would be that while motifs, as referred to above, are intended as patterns on interconnections that may encode functional properties of a given network, partitioning techniques would find community structures which are not granted to match the topologies that one could find with a motif-based approach. Moreover, a definition of sub-graph homogeneity would be necessary. Some previous work has been devoted to such definitions—in Reference [41], the (information-theoretic) entropy of a connections group is used as a measure of homogeneity, while in Reference [42] internal coherence metrics are adopted. One subtlety to account for is that network motifs are not granted to be perfectly homogeneous, hence topologies in which some edges are negative-valued and others are positive-valued could be useful. However, this approach could significantly speed up the sub-graphs research process, so it would constitute an interesting topic for future research.

## Figures and Tables

**Figure 1 entropy-22-00204-f001:**
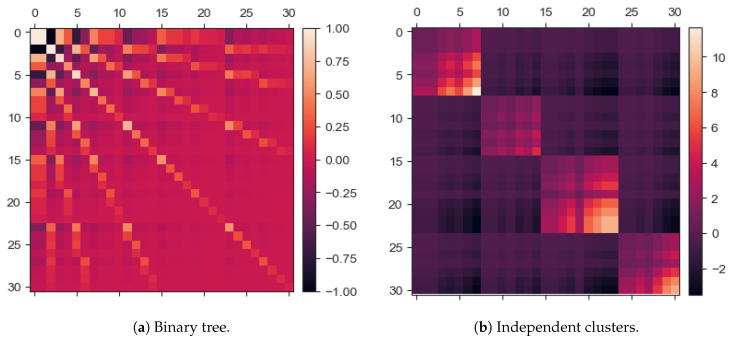
Covariance matrices of the two data sets considered. In (**a**) the variables involved in the covariance computation are all the nodes of the tree structure, from the root node to the leaves. In (**b**) the variables involved are all those constituting the graph in Figure 3. Note that in covariance matrices it is not granted that the elements range in [−1,+1]. In fact, the features of the tree-generated data attain values among {−1,+1}, while in the clusters data set, the random variables involved in the covariance computation attain their topological ordering values. The labels of the features range between 0 and 30, meaning that there are 31 features overall, in both the data sets.

**Figure 2 entropy-22-00204-f002:**
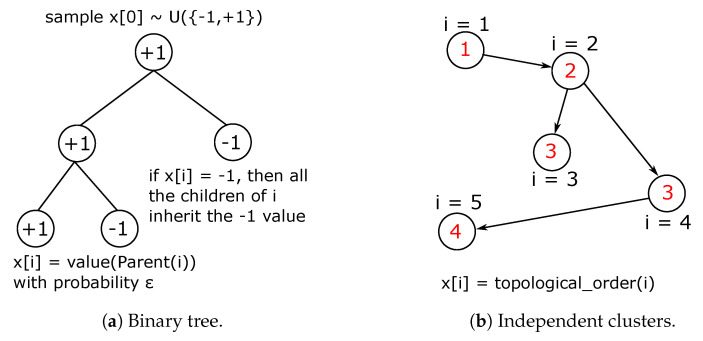
Rationale behind the data sets creation. Note that in the clusters case the red labels represent the topological orderings of the respective nodes. The *i* indices represent the nodes numbers.

**Figure 3 entropy-22-00204-f003:**
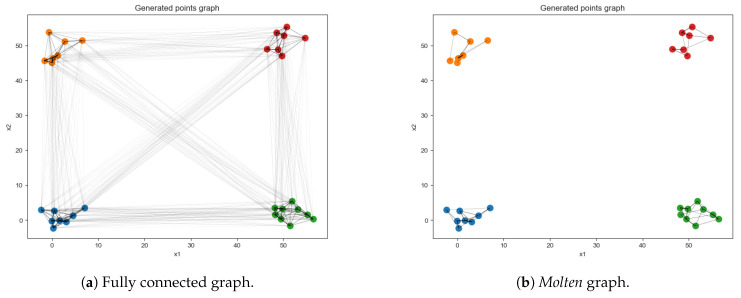
Representation of the process for generating data instances from the independent clusters, showing two subsequent stages of the data set generation: The connections between different groups are gradually eliminated in order to obtain independent graphs. Note that the geometric coordinates do not impact the values attained by the nodes; they are temporarily assigned during the creation stage for the purpose of visualization.

**Figure 4 entropy-22-00204-f004:**
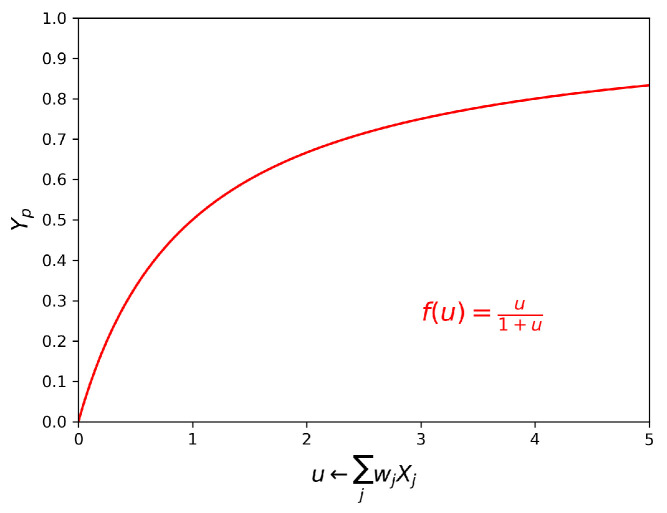
Behavior of the threshold function which quantifies the activity of target kinase, that is $Y_{p}$, as a function of the weighted sum of the input signals. A sensible value of the input weighted sum for the target unit to show activity is assumed to be approximatively 1 [9]. Would one not to make such an assumption, then the expression of the hyper-locus referred to in the main text is more generally $\sum_{j}w_{j}X_{j}=k$.

**Figure 5 entropy-22-00204-f005:**
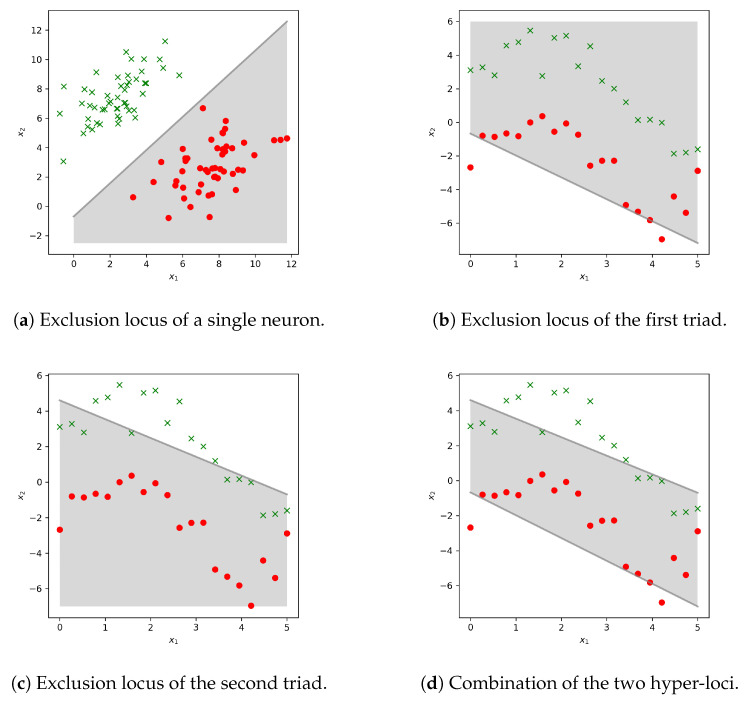
In (**a**) the $x_{1}$ and $x_{2}$ coordinates represent the features of a fictitious data vector, featuring two random variables, in a case of linear separability. Here two input neurons map the input features to a binary label. In (**d**), stacking exclusion hyèer-loci as those in (**b**,**c**), due to a single neuron, one can obtain more intricate decision boundaries. In this graph, it is shown how the joint contribution of two such loci can allow one to go beyond the case of binary classification and linear separable classes, once the problem becomes more complex.

**Figure 6 entropy-22-00204-f006:**
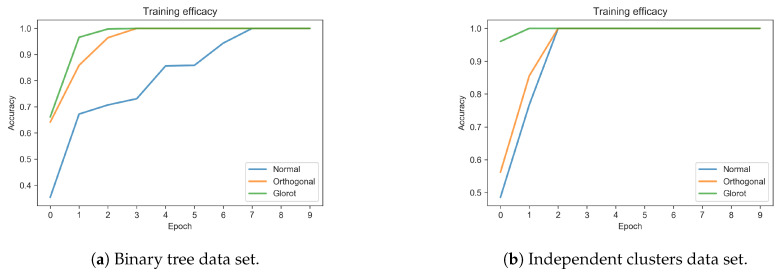
Efficacy of initialisation schemes for (**a**) binary tree and (**b**) independent clusters data sets. Note that the orthogonal matrices initialisation grants the best performance in terms of training speed and note also that the independent clusters environment is easier to be learned, likely owing to its statistical sparsity.

**Figure 7 entropy-22-00204-f007:**
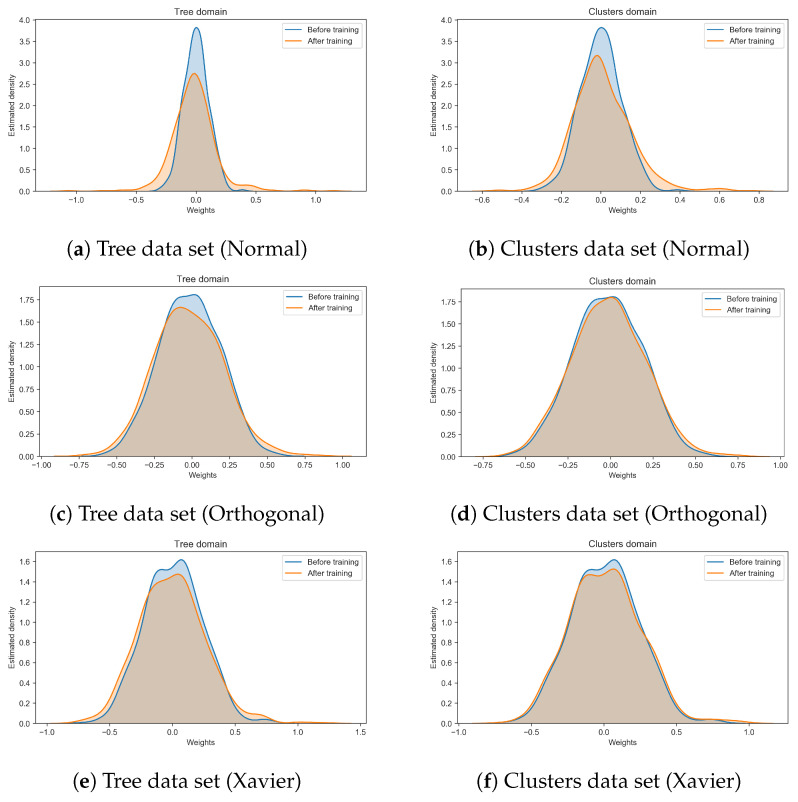
These plots depict the variation that the weights population experiences when trained on different data sets, using different initialization strategies. Results refer to the network 240120.

**Figure 8 entropy-22-00204-f008:**
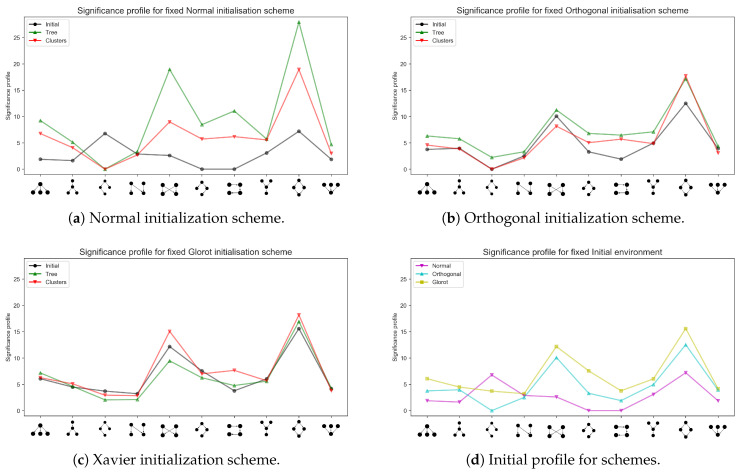
Four-nodes motifs. Significance profiles accounting for different initialization schemes and the case of the initial landscape for different initialization schemes. Note that in panel a, owing to the small variance, the initial significance profile is flatter. In panel d the profiles depict the fingerprint each initialization scheme impresses to the initial significance landscape, that is, curves therein are the collection of the black curves in the first three panels, that refer to the initial significance profile. The Normal initialization scheme is clearly milder than the other two, due to the values sampled by each initial conditions generation. Note that the seventh motif (from the left) is a chain involving one node of all the four layers: it is displayed folded for graphical convenience. Results refer to network 240120.

**Figure 9 entropy-22-00204-f009:**
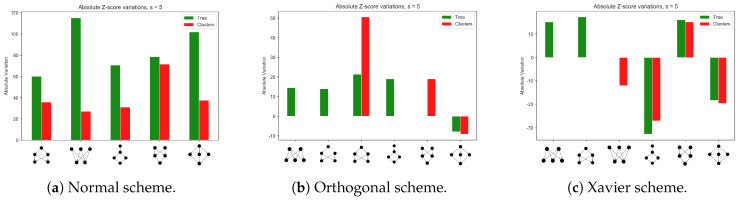
Five-nodes motifs. Total *Z*-score variations accounting for the difference in significance before and after training. Figure refers to most significant motifs, having analysed the weighted graph from the model. Results refer to network 240120.

**Table 1 entropy-22-00204-t001:** Network architectures tested. Note that the name of the network is the seed value used for reproducibility purposes. ReLU stands for Rectified Linear Unit, see Reference [10].

Network	Layer	Units	Activation
240120	Input	31	–
	Hidden 1	20	ReLU
	Hidden 2	10	ReLU
	Output	4	Sotfmax
250120	Input	31	–
	Hidden 1	20	ReLU
	Hidden 2	20	ReLU
	Output	4	Sotfmax
180112	Input	31	–
	Hidden 1	30	ReLU
	Hidden 2	30	ReLU
	Output	4	Sotfmax

**Table 2 entropy-22-00204-t002:** Mean and standard deviation of the weights absolute values, before (i.e., “initial”) and after learning on the two different data sets. This analysis was carried out on the set of non-negligible weights, as explained in Section 2.5.

Initialization	Initial	Tree	Clusters
Normal	0.206±0.206	0.283±0.319	0.254±0.265
Orthogonal	0.346±0.358	0.370±0.400	0.366±0.380
Glorot	0.376±0.387	0.410±0.432	0.390±0.404

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
