# Peer review of "Emergence of Network Motifs in Deep Neural Networks"

_entropy, 2020, doi:10.3390/e22020204_

Round 1

Reviewer 1 Report

The paper analyzes the emergence of "network motifs" in a neural network trained to classify two artificial datasets. They show that the two most prevalent motifs in the network are similar to those found in a class of biological networks called "transduction networks".

While the paper goes quite in depth into the dataset generation process, not the same attention is devoted to the construction of the network and its optimization. In particular, I have a few general comments that I would like to be investigated (or properly answered) in a revised version.

1) First of all, the authors are analyzed simple fully-connected networks, by minimizing the log likelihood of the data (cross-entropy). I assume that motifs are searched by excluding weights that went to zero? However, there is nothing in the optimization that promotes sparsity, so many redundant or small weights are kept in this setup.

2) How does smaller neural networks fare in this setup? It is known that for simple problems, only the output layer of the network can be adapted. In this case, the experiments performed here would not be very meaningful.

3) Connected to point (2), how much does the network (zero / non-zero weights) changes compared to the beginning? Are the motifs simply related to the way the network is built?

4) Similarly, some analysis of the robustness of the conclusions w.r.t. the architecture should be provided: what happens if we vary the number of units or the number of layers? What happens if we try on a real-world dataset?

Language also needs corrections. There are many poorly types sentences such as "Hence different of such strategies are tested". Caption of Fig. 1 and Footnote 3 have sentences that never close.

Author Response

We are grateful to the Reviewer for the precious feedback. We try to address all the concerns in the answers below, which will also explain how the manuscript has been improved accordingly.

Comment 1. We agree with the Reviewer that the inclusion of further constraints on the learning process (e.g., sparsity or weight regularization) could have an impact on the emergent motifs, since weights would be pushed toward zero. We avoided including these additional optimization constraints in order to more neatly investigate the ``pure'' effect of stochastic gradient descent on the final network topology. However, we note that besides excluding weights that went to zero, we also excluded weights that were relatively close to zero, in order to only detect relevant motifs. To this aim, we identified two boundary values in the weights distribution, simmetrically placed with respect to the mean (which is indeed close to zero), that defined the range of values to be excluded from the analysis (see Figure C.1 in the Appendix C). Of course the choice of the threshold used to define such interval is critical, as is the choice of the thresholds used to define the intervals for the discretization of weights into the mildly positive and mildly negative categories (as described in the Appendix). We now further elaborate on these issues in the concluding section of the revised manuscript.

Comment 2. This is an interesting observation, since our data set are indeed made of data patterns that might be classified even by networks without hidden layers. However, in the spirit of the previous work (e.g., Saxe et al. 2013) we aimed at understanding how deep networks could self-organize in non-trivial ways even when faced with relatively simple classification tasks, which also allowed for a better control of the statistical signature contained in the different data sets. Notably, as pointed out by Alon (2007), in certain biological multi-layered networks (transcription networks, which are not the transduction networks discussed in our manuscript) it is the final layer to exhibit the most meaningful structures. In our model we similarly observed that the largest weights variations indeed occur in the final layer; we added a short discussion about these issues in the revised version of the manuscript.

Comment 3. The weights change could be assessed by measuring the difference of the estimated densities. We performed such analysis, and found that the learning process indeed changes the weights distribution. Interestingly, it turns out that this depends on the initialization scheme adopted: In the case of the Normal distribution initialization, the initial values sampled are smaller and exhibit larger variations. We report this additional analysis in the revised version of the manuscript. Regarding the second question: since the initial architecture is fixed (and bipartite across layers), some motifs (see Figure 5 in the main text) could indeed be favored. A better assessment of this phenomenon requires a deeper mathematical analysis on the combinatorial probability of such wirings, which is beyond the scope of the present paper. However, we now more explicitly discuss this point in the manuscript.

Comment 4. Scaling up the deep network requires an exponential increase in computational time for the motifs mining software, which prevents a systematic exploration of this problem in the present study. Nevertheless, we added a new set of simulations, reported in the Appendix, which show that our results hold when changing the number of units in the hidden layers. We also trained and analysed one deeper architecture on a more realistic problem (handwritten digit recognition), and found that the results are well aligned with those emerging from the artificial data sets. We now report also these findings in the Appendix.

Comment 5. We performed an accurate proof-correction of language and typos in the revised version of the manuscript, and we fixed the caption of Fig. 1 and Footnote 3.

References.

Uri Alon ``Network motifs: theory and experimental approaches''. In: Nature Reviews Genetics 8.6 (2007), pp. 450–461. ISSN: 1471-0064. DOI: 10 . 1038 /
nrg2102.

Andrew M. Saxe, James L. McClelland, and Surya Ganguli. Exact solutions to
the nonlinear dynamics of learning in deep linear neural networks. 2013. arXiv:
1312.6120 [cs.NE].

Reviewer 2 Report

In this paper, authors have studied emergence of network motifs as different functional topological modules.
Authors have carried out simulations which show that final network topology is primarily shaped by learning dynamics.

However, it was also observed that the final network topology can be strongly biased by choosing appropriate weight initialization schemes.
  Authors suggest initialization strategies can make learning more effective by promoting the development of useful network motifs.   Authors already have provided a brief literature review, but a comparison of the existing approaches is limited.   I suggest that authors also perform a more detailed and comprehensive comparative analysis of the related works on analyzing network motifs, in which they compare each existing related studies with each other against a criteria based on different comparison factors, e.g., efficiency, time taken, resource consumption, computation consumption, accuracy, precision, and etc.   Authors should discuss complexity analysis of the different proposed approaches too. For example, what is the complexity and cost of the algorithms with respect to time, space and how does it increase or decrease and under what circumstances?   Authors should also discuss any possible limitations of their proposed approaches for simulations. What is the scalability measure? And based on what factors does it vary?

Author Response

Comment 1. I suggest that authors also perform a more detailed and comprehensive comparative analysis of the related works on analyzing network motifs, in which they compare each existing related studies with each other against a criteria based on different comparison factors, e.g., efficiency, time taken, resource consumption, computation consumption, accuracy, precision, and etc. Authors should discuss complexity analysis of the different proposed approaches too. For example, what is the complexity and cost of the algorithms with respect to time, space and how does it increase or decrease and under what circumstances?

Answer. We note that a broad analysis on the algorithmic details underlying the motifs mining software used, and more generally the network motifs algorithms, is out of the scope of the present work. Here our focus is rather on how to apply such tools to gain some understanding of computational properties of deep networks. In the revised version of the manuscript we now mention one recent review paper in which further details about these aspects are provided, see Masoudi-Nejad (2012).

Comment 2. Authors should also discuss any possible limitations of their proposed approaches for simulations. What is the scalability measure? And based on what factors does it vary?

Answer. While the issues above lie outside the scope of the present paper, we agree with the Reviewer that limitations due to scalability issues should be at least mentioned in our discussion. To this aim, we now included a brief discussion about computational complexity of the motifs mining software in the final section of the manuscript.

References.

Ali Masoudi-Nejad, Falk Schreiber, and Zahra R.M. Kashani. “Building blocks
of biological networks: a review on major network motif discovery algorithms”.
In: IET Systems Biology 6.5 (Oct. 2012), pp. 164–174. DOI: 10.1049/ietsyb.
2011.0011. URL: https://doi.org/10.1049/iet- syb.
2011.0011.

Reviewer 3 Report

This is an interesting paper. The paper provides incidences of how network science tools are successful for the study of artificial neural networks. 

The following questions are for the authors. 

Can the proposed network topology used for deep learning methods, such as CNNS or LSTM or other types of network. A comprehend review is depicted in [1] How the classification performance affecting on the parameters of the method adopted? Spectal graph clustering which is a way for determining subgraphs within a graph in a way that intra coherency is optimised an inter coherency is minimised can be considered to have a relationship with the proposed method [2].  

Refs.

[1] Voulodimos, Athanasios, et al. "Deep Learning for Computer Vision: A Brief Review." (2018).

[2] J. Shi and J. Malik, “Normalized Cut and Image Segmentation,” IEEE Trans. Pattern Analysis and Machine Intelligence, vol. 22, no. 8, pp. 888-905, Aug. 2000.

Author Response

Comment 1. Can the proposed network topology used for deep learning methods, such as CNNS or LSTM or other types of network. A comprehend review is depicted in Voulodimos et al. (2018).

Answer. The proposed approach could indeed in principle work for any kind of model which can be formalized as a graph, or could be rendered in a similar amenable format. In the present case, the choice of the model fell on the multi-layer perceptron for its explicit graph appearance. We added a short discussion about this in the revised version of the manuscript, also mentioning the review paper indicated by the Reviewer.

Comment 2. How the classification performance affecting on the parameters of the method adopted?

Answer. If sufficiently trained on our data sets of synthetic data, all models peak to the top accuracy value with both the binary tree and the independent clusters data set. However, it is interesting to note that the efficacy of learning (i.e., convergence speed) is indeed influenced by some parameters, most notably, by the initialization scheme. As discussed in the main text (and reported in Fig. 4), orthogonal matrices (see Saxe et al., 2013) or Xavier (see Glorot and Bengio, 2010) initializations favor a faster learning. Another quantity that impacts learning speed is the batch size, the amount of training samples the model is exposed to in each epoch of training. The smaller the batch size, the faster the learning stage.

Comment 3. Spectal graph clustering which is a way for determining subgraphs within a graph in a way that intra coherency is optimised and inter coherency is minimised can be considered to have a relationship with the proposed method (Shi, 2000).

Answer. This is indeed an interesting observation. The main difference we see is that network motifs are intended as recurrent patterns of interconnections between nodes, not only as sub-graphs in which the network could be partitioned into. Also, the definition of a measure of homogeneity to identify coherent groups of connections would be necessary. Some work has been devoted to this point (see in particular Choodbar et al. (2012) and Onnela et al., 2005). In this work, the authors depict ideas to mine motifs in weighted networks, such as by evaluating the entropy of an edges group (intended as information entropy) and internal coherence. We thus agree that comparing the results of motifs mining with spectral approaches could be a topic for further research work, as we now discuss in the revised version of the manuscript.

References.

Athanasios Voulodimos et al. “Deep Learning for Computer Vision: A Brief Review”. In: Computational Intelligence and Neuroscience 2018 (2018). Article.
ISSN: 16875265.

Andrew M. Saxe, James L. McClelland, and Surya Ganguli. Exact solutions to
the nonlinear dynamics of learning in deep linear neural networks. 2013. arXiv:
1312.6120 [cs.NE].

Xavier Glorot and Yoshua Bengio. “Understanding the difficulty of training deep
feedforward neural networks”. In: Proceedings of the Thirteenth International
Conference on Artificial Intelligence and Statistics. Ed. by Yee Whye Teh and
Mike Titterington. Vol. 9. Proceedings of Machine Learning Research. Chia Laguna Resort, Sardinia, Italy: PMLR, 2010, pp. 249–256.

Sarvenaz Choobdar, Pedro Ribeiro, and Fernando Silva. “Motif Mining inWeighted
Networks”. In: 2nd IEEE ICDM Workshop on Data Mining in Networks. IEEE,
2012, pp. 210–217. DOI: 10.1109/ICDMW.2012.111.

Jukka-Pekka Onnela et al. “Intensity and coherence of motifs in weighted complex networks”. In: Phys. Rev. E 71 (6 2005). DOI: 10.1103/PhysRevE.71.
065103.

Round 2

Reviewer 3 Report

All my previous comments have beed addressed.